# Loss of *Runx1* Induces Granulosa Cell Defects and Development of Ovarian Tumors in the Mouse

**DOI:** 10.3390/ijms232214442

**Published:** 2022-11-21

**Authors:** Kamiya Bridges, Humphrey H.-C. Yao, Barbara Nicol

**Affiliations:** Reproductive and Developmental Biology Laboratory, National Institute of Environmental Health Sciences, Research Triangle Park, NC 27709, USA

**Keywords:** RUNX1, ovary, granulosa cells, tumor, cancer

## Abstract

Genetic alterations of the *RUNX1* gene are associated with a variety of malignancies, including female-related cancers. The role of *RUNX1* as either a tumor suppressor gene or an oncogene is tissue-dependent and varies based on the cancer type. Both the amplification and deletion of the *RUNX1* gene have been associated with ovarian cancer in humans. In this study, we investigated the effects of *Runx1* loss on ovarian pathogenesis in mice. A conditional loss of *Runx1* in the somatic cells of the ovary led to an increased prevalence of ovarian tumors in aged mice. By the age of 15 months, 27% of *Runx1* knockout (KO) females developed ovarian tumors that presented characteristics of granulosa cell tumors. While ovaries from young adult mice did not display tumors, they all contained abnormal follicle-like lesions. The granulosa cells composing these follicle-like lesions were quiescent, displayed defects in differentiation and were organized in a rosette-like pattern. The RNA-sequencing analysis further revealed differentially expressed genes in *Runx1* KO ovaries, including genes involved in metaplasia, ovarian cancer, epithelial cell development, tight junctions, cell−cell adhesion, and the Wnt/beta-catenin pathway. Together, this study showed that *Runx1* is required for normal granulosa cell differentiation and prevention of ovarian tumor development in mice.

## 1. Introduction

Ovarian cancer, although not the most common cancer, represents the deadliest gynecological disease. There are several types of ovarian tumors that relate to their histological features and cell origins. The three main types include epithelial tumors, sex-cord stromal tumors, and germ-cell tumors [1]. Among these, epithelial tumors arising from either ovarian surface epithelial cells or fallopian tubes are the most frequent type of ovarian cancer, accounting for approximately 90% of cases. Sex-cord stromal tumors, although much less frequent (roughly 7%), can develop from stromal cells (thecoma) or the sex cords (granulosa cell tumors or GCTs) [2]. Due to the lack of efficient screening exams as well as the asymptomatic nature of the disease, ovarian cancer is usually not detected, until it reaches an advanced stage. This exposes a need for more understanding regarding the pathogenesis of the disease.

Ovarian tumors can stem from exposure to environmental agents, to genetic factors, or both. Runt-related transcription factor 1 (RUNX1) is a transcription factor that belongs to the RUNX transcription factor family, along with RUNX2 and RUNX3 [3]. Through its capacity to directly activate or suppress the expression of targets genes, RUNX1 plays key roles in early development and cell lineage specification in various tissues [4]. Consequently, mis-regulation of *RUNX1* is associated with cancers. For instance, RUNX1 is required for normal hematopoiesis [5], and genetic alterations of the *RUNX1* gene have been linked to various forms of leukemia and other hematological malignancies [6]. While *RUNX1* mis-regulation has been originally explored in blood-related cancers, its role has now been identified in many other types of cancers, particularly tumors of epithelial origins [7]. Depending on the type of cancer, *RUNX1* can act as either an oncogene or a tumor suppressor. Among the different types of cancers, mis-regulation of *RUNX1* is associated with female-related cancers such as breast cancer, uterine cancer, and ovarian cancer [8,9]. In particular, genetic alterations of the *RUNX1* gene, including the amplification and deep deletion, have been previously documented in 1.5% of ovarian cancers according to the public data from The Cancer Genome Atlas (TCGA).

Although mis-expression of *RUNX1* is associated with ovarian cancer, a direct causal link of *Runx1* to the cancer formation is lacking. In the mouse, *Runx1* is involved in early ovarian development and is expressed in both granulosa cells and the ovarian surface epithelium in the fetal ovary [10]. In this study, we investigated whether loss of *Runx1* affected the mouse adult ovary and could cause the development of ovarian pathologies and tumors. To investigate the potential role of *Runx1* in ovarian tumor development, we developed a conditional knockout (KO) of *Runx1* in the somatic cells of the ovary. We found that loss of *Runx1* expression led to ovarian defects and increased risk of developing ovarian tumors in aged mice.

## 2. Results

### 2.1. Runx1 Is Expressed in Granulosa Cells and the Surface Epithelium of Adult Mouse Ovaries

Expression of RUNX1 in the ovary begins at the fetal stage. We previously showed that RUNX1 is expressed in granulosa cells and the surface epithelium of the mouse fetal ovary [10]. To determine if it was also the case in the adult ovary, we used a reporter mouse model that expresses enhanced green fluorescent protein (EGFP) under the control of *Runx1* promoter (Figure 1A) [11]. *Runx1*-EGFP expression was detected in both ovarian surface epithelium and granulosa cells, overlapping with the known granulosa cell marker Forkhead Box L2 (FOXL2) [12]. Ovarian surface epithelium and granulosa cells are the source of two different types of ovarian cancers [1]. Therefore, to determine whether loss of *Runx1* in these cell populations could lead to these types of ovarian cancer, we generated a conditional knockout mouse model, in which *Runx1* was specifically ablated from the somatic cells of the ovary (Figure 1B). For this purpose, we used a model that expressed Cre under the control of nuclear receptor subfamily 5, group A, member 1 (*Nr5a1*) encoding for Steroidogenic Factor 1 (SF-1). This *Nr5a1*-Cre allows the targeting of all ovarian somatic cells, including granulosa cells and surface epithelium cells [13]. The resulting *Runx1* conditional knockout mouse model had a mixed B6D2 and B6.129 genetic background. We confirmed that *Runx1* was ablated successfully in *Runx1* KO adult ovaries by RT-PCR (Figure 1C).

### 2.2. Loss of Runx1 Leads to Increased Prevalence of Ovarian Tumors

To assess the effects of *Runx1* conditional knockout, we examined ovaries of control and *Runx1* KO mice at 15 months (Figure 2A). The majority of both control and *Runx1* KO ovaries appeared macroscopically normal at the time of collection. However, *Runx1* KO ovaries were significantly heavier than control ovaries (Appendix A). In addition, there was a higher occurrence of macroscopic tumors on *Runx1* KO mice. Macroscopic tumors were observed in 3.7% of control females (1 out of 27 females) and in 27.3% of *Runx1* KO mice (9 out of 33 females) (Figure 2B). These tumors were bilateral for 44% of these *Runx1* KO mice and unilateral in the one control female (Figure 2B). Hematoxylin and eosin staining was used to analyze the histology of the ovaries collected at 15 months (Figure 2C). Control ovaries exhibited a typical ovarian histology for that age, with few follicles left (Figure 2C arrows) and some corpora lutea (Figure 2C). The 15-month *Runx1* KO ovaries that did not have macroscopic tumors at the time of collection presented dense follicle-like lesions composed of cells with a scant amount of cytoplasm arranged in rosette-like pattern (Figure 2C and Appendix A). *Runx1* KO ovaries also developed a variety of histological anomalies, such as presence of cysts, hemorrhagic follicles, and anomalies of the ovarian surface epithelium (Figure 2C and Appendix A). *Runx1* KO tumors presented a diffuse growth of relatively uniform cells organized in sheets, with small eosinophilic cavities and hemorrhagic cysts, reminiscent of granulosa cell tumors (Figure 2C lower panel and Appendix A). To identify the cell type composing the tumors, sections of 15-month ovaries were stained for Keratin-8 (KRT8), a marker for the surface epithelium, and FOXL2, a marker for the granulosa cells (Figure 2D). In control ovaries, positive labeling for FOXL2 was observed in the follicles and corpora lutea (Figure 2D). On the other hand, KRT8 was strongly expressed in the ovarian surface epithelium and weakly detected in small secondary follicles as previously shown [14]. In *Runx1* KO mice, cells composing the tumors showed an intense expression for FOXL2, indicating that the tumors were granulosa cell tumors.

### 2.3. Young Runx1 KO Mice Present Ovarian Defects but No Apparent Tumors

Next, we examined ovaries at an earlier age (4.5 months) to determine if they already presented some ovarian defects (Figure 3). In control ovaries, follicles were composed of an oocyte surrounded by symmetrical homogenous layers of granulosa cells (Figure 3A). *Runx1* KO ovaries at 4.5 months contained follicles at various stages similar to the control ovary, and no macroscopic masses were observed (Figure 3B). Histological analyses revealed the presence of normal follicles, corpora lutea, and dense follicle-like lesions in all *Runx1* KO ovaries (Figure 3B). These follicle-like lesions were similar to those found at 15 months, composed of dense cells organized in a rosette-like pattern. Along with these aberrant follicles, cysts were also observed in the *Runx1* KO ovaries (Figure 3B).

### 2.4. The Follicle-like Lesions in Runx1 KO Ovaries Are Composed of Quiescent Abnormal Granulosa Cells

To further assess the characteristics of the follicle-like lesions in 4.5-month-old ovaries, we performed immunostaining for the granulosa cell marker FOXL2 and the proliferation marker KI-67 and detection of cell death using the Terminal deoxynucleotidyl transferase dUTP nick end labeling (TUNEL) assay. Control ovaries contained growing follicles with FOXL2+/KI-67+ proliferating granulosa cells and some atretic follicles with TUNEL+ dying granulosa cells (Figure 4A). In *Runx1* KO ovaries, some normal growing follicles and atretic follicles were also observed (Figure 4A). The follicle-like lesions were FOXL2+ (Figure 4A), confirming that they arose from ovarian follicles. These aberrant follicles were devoid of oocytes, replaced by an empty eosinophilic cavity (Figure 3B insets and Figure 4A). Considering that many of these aberrant follicles showed an asymmetrical organization of FOXL2+ cells with only one or two cell layers on one side, we suspected that these aberrant follicles arose from primary or early secondary follicles (Figure 4A inset). These abnormal granulosa cells were negative for both KI-67 and TUNEL, indicating that they were not proliferating as normal follicular cells do nor were they undergoing cell death. Anti-Müllerian hormone (AMH), a marker for granulosa cells of primary to early antral follicles, was detected in normal secondary follicles of control and *Runx1* KO ovaries (Figure 4B). However, granulosa cells composing the follicle-like lesions in *Runx1* KO ovaries failed to express AMH (Figure 4B), indicating defects in granulosa cell differentiation at the primary follicular stage. Immunofluorescence for laminin revealed that control ovary follicles were fully surrounded by a single basal membrane composed of laminin (Figure 4C). In contrast, the follicle-like lesions in *Runx1* KO ovaries were not fully surrounded by a single basal membrane but instead showed laminin deposition around each rosette structures composing these lesions (Figure 4C inset). Overall, the absence of cell proliferation, AMH expression and proper basal membrane deposition suggested some defects in the differentiation and function of the granulosa cells in these abnormal follicle-like lesions in *Runx1* KO ovaries. Finally, in older mice (12–15 months), a subset of these FOXL2+ follicle-like lesions also expressed KRT8, a marker usually found in the epithelium and weakly detected in granulosa cells of small secondary follicles (Figure 4D). Expression of KRT8 or other keratins was also observed in subsets of abnormal granulosa cells in other mouse models for granulosa cell tumors [14,15] and in KGN cells, a cell line originating from a human adult granulosa cell tumor [16]. These findings together indicated that these abnormal granulosa cells acquired some epithelial-like characteristics.

### 2.5. Differentially Expressed Genes in Runx1 KO Ovaries Are Associated with Epithelium Development, Tight Junctions, and Metaplasia

To gain more insights into the early consequences of *Runx1* loss in adult ovaries, we performed RNA-seq on both *Runx1* KO and control ovaries at 4.5 months of age (*n* = 6/genotype). Gene expression analysis identified 349 genes that were differentially expressed (DEG) between the two groups (Figure 5A; Appendix A). Among these 349 genes, 164 were upregulated and 185 were downregulated in *Runx1* KO ovaries. Gene ontology (GO) analyses revealed that the top mis-regulated GO biological processes were related to epithelial cell development/morphogenesis, tight junctions, and cell-cell adhesion (Figure 5B). For instance, there was significant downregulation of tight junction Claudin (*Cldn*) genes *Cldn3/4/7* (Figure 5E and Appendix A). Tight junctions are important for cell adhesion, polarity, growth, migration, and paracellular permeability, and their destabilization is associated with malignancy [17]. The Kyoto Encyclopedia of Genes and Genomes (KEGG) pathway analysis also revealed mis-regulation of tight junctions and cell adhesion (Figure 5C). It is unclear whether mis-regulation of genes in epithelial cell development/morphogenesis and tight junctions involves the surface epithelium itself or the granulosa cells, as adult granulosa cells originate from the ovarian surface epithelium during neonatal development of the ovary [18,19]. It is, however, worth noting that aged *Runx1* KO ovaries displayed irregular ovarian surface epithelium with some invaginations (Appendix A).

Next, we used DisGeNet, a platform integrating information on human disease-associated genes and variants [20], to identify the pathways linked to human diseases. We found an association of differentially expressed genes with metaplasia (Figure 5D). The differentially expressed genes were also associated with formation of cysts, pseudohermaphroditism, as well as malignant neoplasm of the ovary, epithelial ovarian cancer, and ovarian carcinoma (Figure 5D). Among the differentially expressed gene that were associated with ovarian cancers, *Cdh1*, encoding E-Cadherin, was strongly downregulated in *Runx1* KO ovaries. Downregulation of *Cdh1* impairs epithelial cell adhesion and is associated with ovarian epithelial cancer [21,22]. On the other hand, other genes coding for other cadherins such as *Cdh13* and *Cdh23* were significantly upregulated. Poly [ADP-ribose] polymerase 1 (*Parp1*), a DNA damage repair enzyme significantly increased in several malignant tissues [23], was upregulated in *Runx1* KO ovaries (Figure 5E and Appendix A). Finally, *Amh* was downregulated in *Runx1* KO ovaries, consistent with the lack of AMH protein in follicle-like lesions (Figure 4B). On the other hand, *Foxl2* expression was not significantly changed (Figure 5E and Appendix A).

Other than the tight junction and cell adhesion genes, the Wnt signaling pathway, which is involved in granulosa cell tumor development in mice [15,24], was also significantly mis-regulated (Figure 5C and Figure 6A and Appendix A). For instance, the WNT receptor Frizzled-10 (*Fzd10*) was upregulated and was found upregulated in human granulosa cell tumors [25] and colorectal cancer [26]. Secreted Frizzled Related Protein 2 (*Sfrp2*), which acts as both an activator or a repressor of the Wnt/β-catenin pathway [27], was also significantly upregulated in the *Runx1* KO ovaries (Figure 6A). Finally, R-spondin1 (*Rspo1*), a potentiator of the canonical Wnt/β-catenin pathway in granulosa cell differentiation and granulosa cell tumor development [15,28], was upregulated in the *Runx1* KO ovaries (Figure 6A). To determine what cell population expressed *Rspo1*, we performed RNA-scope for *Rspo1* transcript in 4.5-month-old control and *Runx1* KO ovaries (Figure 6B). *Rspo1* was detected in small and secondary follicles in both control and Runx1 KO ovaries. In *Runx1* KO ovaries, *Rspo1* was also detected in a subset of follicle-like lesions (arrowheads), while some follicle-like lesions presented little to no *Rspo1* expression (arrows). Overexpression of *Rspo1* in the mouse ovary has been associated with the development of granulosa cell tumors [15].

Finally, we compared the differentially expressed genes (DEGs) in our model with the published dataset for TGFBR1-CA ovaries, a mouse model of granulosa cell tumors caused by constitutively active TGFβ signaling [29]. There was little overlap among the genes differentially expressed in the two models (Appendix A). Among the 349 DEGs in *Runx1* KO, only 14 DEGs were co-upregulated, and 6 DEGs were co-downregulated with the TGFBR1-CA model. For instance, both models presented a significant upregulation of genes involved in extra-cellular matrix remodeling, such as Matrix metalloproteinase-9 (*Mmp9*) and Fibroblast Growth Factor 7 (*Fgf7*), and an upregulation of Cadherin 6 (*Cdh6*), involved in cell−cell adhesion. Such little overlap among the DEGs suggested that different signaling pathways are involved in these two mouse models.

## 3. Discussion

RUNX1 has been associated with various types of cancers, particularly hematological malignancies and cancers of epithelial origins. Here, we showed that in the mouse adult ovary, *Runx1* was expressed in the supporting granulosa cells and the ovarian surface epithelium. These two cell populations are the source of two different types of ovarian cancers both in humans and in mouse models. Therefore, we suspected that *Runx1* misexpression could result in either or both ovarian tumor types. Using a conditional knockout of *Runx1* in the somatic cells of the ovary, we discovered that *Runx1* loss led to abnormal follicle-like lesions in young mice and appearance of ovarian tumors in aged mice. The follicle-like lesions consisted of quiescent FOXL2+ granulosa cells that improperly differentiated, failing to express the granulosa cell marker AMH that is normally expressed in granulosa cells from the primary to pre-antral follicular stage. The morphology of these granulosa cells, with a rosette-like pattern of granulosa cells capable of developing their own basement membrane [30], resembled that of granulosa cell tumors (GCT). By the age of 15 months, 27% of *Runx1* KO mice presented macroscopic ovarian tumors composed of FOXL2+ granulosa cells. Altogether, our findings showed that loss of *Runx1* in the mouse ovarian somatic cells increased the incidence of granulosa cell tumors.

There are multiple mouse models that develop these asymmetrical quiescent follicle-like lesions, such as the activation of the Wnt/β-catenin pathway [24], overexpression of *Rspo1* [15], Forkhead Box O1/O3 (*Foxo1*/3) double knockout [14], and mis-regulation of the RAS-ERK1/2 signaling pathway [31,32]. Most of these models eventually develop granulosa cell tumors. Depending on the signaling pathway altered, these models present a broad range of tumorigenesis penetrance (10% to 100%) and age of tumor appearance (from a few weeks to more than a year). It remains unclear how these quiescent lesions in younger ovaries eventually give rise to large tumors later on. While 100% of *Runx1* KO ovaries contained follicle-like lesions early on, only 27% of them developed macroscopic tumors at an advanced age. This suggested that another genetic event might be required for the transition from quiescent abnormal granulosa cells to tumors.

RNA-sequencing analysis of 4.5-month-old *Runx1* KO ovaries revealed differential expression of genes associated with ovarian cancer. For instance, *Rspo1*, a potentiator of the Wnt/β-catenin pathway in granulosa cells [28], was upregulated. Notably, a mouse model that overexpresses *Rspo1* in the ovary leads to similar phenotype as seen in our *Runx1* KO mouse model [15]. Ovaries overexpressing *Rspo1* also contain follicle-like lesions composed of quiescent granulosa cells that organize in a rosette-like pattern. Similar to our model, these abnormal granulosa cells fail to express AMH and present epithelial characteristics with the expression of keratins and altered intercellular junctions. Ovaries overexpressing *Rspo1* also developed a late onset of macroscopic tumors in 10–12% of 12-month-old females. This study concluded that ectopic RSPO1 activation in adult ovaries likely induced formation of GCTs by regulating not only canonical Wnt/β-catenin signaling activation, but also intercellular junction homeostasis in granulosa cells. The similarities between the two mouse models support the idea that the ovarian phenotypes in *Runx1* KO mice is linked to mis-regulation of Wnt/β-catenin signaling and/or cell junctions through the upregulation of *Rspo1*. Previous studies in epithelial cancers revealed an inhibitory relationship between RUNX1 and the Wnt/β-catenin signaling pathway, where *RUNX1* antagonizes the canonical β-catenin signaling. For example, in breast cancers, the loss of RUNX1 in estrogen receptor-positive mammary epithelial cells increases β-catenin signaling and stimulates cell proliferation [33]. Beside the *Rspo1* overexpression mouse model, constitutive activation of β-catenin in the mouse ovary also leads to similar abnormal follicle-like structures, composed of disorganized quiescent granulosa cells, and eventually GCT development [24]. The constitutive activation of β-catenin promotes GCT development much earlier, with a prevalence of 57% at 7.5 months of age, indicating a stronger capacity for tumorigenesis when β-catenin is directly activated. Taken together with our findings and the inhibitory relationship between the Wnt/β-catenin pathway and *Runx1*, we proposed that *Rspo1* upregulation contributes to the development of follicle-like lesions and eventually GCT in aging *Runx1* KO ovaries.

In humans, the main cause of adult GCT (97%) is a point mutation leading to the C134W substitution in the transcription factor FOXL2 [34]. This mutation appears to modify FOXL2 functions as a transcription factor [35,36,37]. Our lab previously demonstrated that transcription factors RUNX1 and FOXL2 play redundant roles in fetal granulosa cell differentiation by directly controlling the expression of common genes [10]. Considering this shared role between FOXL2 and RUNX1 in granulosa cell gene regulation, it is possible that *Runx1* misexpression contributes to granulosa cell tumors as well. RUNX1 is expressed at variable levels in human adult GCTs [16]. It is worth noting that combined comparative genomic hybridization and transcriptomic analyzes of human GCTs revealed that the *RUNX1* gene was recurrently mutated in these tumors [38]. This suggests that, together with *FOXL2* mutation, alterations of the *RUNX1* gene may contribute to GCT development in humans.

While our *Runx1* KO mouse model is associated with GCT development, genomic alterations of *RUNX1* in human ovaries have been so far mostly associated with epithelial ovarian cancer (EOC). *Runx1* KO ovaries did not display tumors of epithelial origins. However, some of the top pathways altered in the transcriptome of young adult *Runx1* KO ovaries were related to epithelium development. While this may reflect the neonatal epithelial origins of granulosa cells or the acquisition of epithelial-like characteristics by abnormal granulosa cells [18], effects of the *Runx1* KO on the surface epithelium homeostasis cannot be excluded. Indeed, aged *Runx1* KO ovaries presented defects on the ovarian surface with hyperplasia and invaginations of the epithelium. Considering the role of *Runx1* in cell differentiation, it is possible that loss of *Runx1* causes some reprogramming in the cells that normally would express *Runx1*, such as granulosa and epithelial cells. It is interesting that in our model and other mouse models of granulosa cell tumors [14,15], some abnormal granulosa cells start expressing epithelial markers. Progenitors for granulosa cells and epithelial cells share a common stemness marker, LGR5. LGR5 is a marker for stem epithelial cells at the surface of the adult ovary and is suspected to play a role in serous ovarian cancer [39,40,41]. In addition, during neo-natal and postnatal ovarian development in the mouse, granulosa cells that are derived from the surface epithelium arise from LGR5+ precursor cells [18,19]. LGR5 is a receptor for RSPO1 and is involved in the Wnt signaling. It would be interesting to further explore the potential role of *Lgr5* in the defects in granulosa cell differentiation and development of follicle-like lesions. The role of RUNX1 in tumorigenesis is complex, acting as an oncogene or a tumor suppressor, depending on the tissue or even on the cell type within the same tissue [8]. The *RUNX1* gene is frequently reported as overexpressed or amplified in different forms of epithelial cancers such as skin and colorectal cancer [7]. While a variety of the *RUNX1* gene alterations have been identified in epithelial ovarian cancer, the most common is amplification [8]. The *RUNX1* gene is overexpressed in tumors from EOC patients [9]. Furthermore, it was found that knockdown of *RUNX1* expression in EOC cells led to strong downregulation in genes involved in cell proliferation and tumorigenesis. Therefore, while RUNX1 seems to play a tumor-suppressor role in mouse granulosa cells in our model, it has a more prominent oncogenic role in human epithelial ovarian cancer. The logical next step is to investigate the effects of *Runx1* overexpression in the mouse ovarian surface epithelium and determine whether it leads to tumorigenesis. The effects of *Runx1* misexpression in epithelial cells of the reproductive tract is also worth looking into based on the fact that many EOC cases actually have an extra-ovarian origin, arising from the fallopian tube epithelium in women [41,42].

In summary, we discovered that *Runx1* is required for normal granulosa cell differentiation and prevention of ovarian tumor development in mice. The conditional knockout of *Runx1* in somatic cells of mouse ovaries leads to abnormal follicle-like lesions and an increased risk of ovarian tumor development in aged mice. These findings suggest a duality between the roles of *Runx1* as a tumor suppressor gene in mouse granulosa cells and as both a tumor suppressor and an oncogene as previously described in human epithelial ovarian cancer.

## 4. Materials and Methods

### 4.1. Mouse Models

Tg(*Runx1*-EGFP) reporter mice were purchased from MMRRC (MMRRC_010771-UCD). *Runx1*^+/−^ (B6.129S-*Runx1^tm1Spe^*/J) [43] and *Runx1^f^*^/*f*^ (B6.129P2-*Runx1^tm1Tani^*/J) [44] mice were purchased from the Jackson Laboratory (stock numbers were 005669 and 008772, respectively). The late Dr. Keith Parker provided *Nr5a1*-Cre^Tg/Tg^ mice (B6D2-Tg(Nr5a1-cre)2Klp). To generate the *Runx1* KO mice (*Nr5a1-Cre^+/Tg^; Runx1^f/^*^−^), *Runx1^f^*^/*f*^ females were crossed with *Nr5a1-Cre^+/Tg^; Runx1^+/−^* males. *Nr5a1-Cre^+/+^; Runx1^+/f^* littermates were considered as controls. 

Mouse female littermates were housed in groups of 2 to 5 in ventilated cages (Techniplast, Exton, PA, USA) under standard 12-h light/dark cycles (6:00–18:00 EST). Autoclaved NIH31 open formula (Harlan Laboratories, Madison, WI, USA) and reverse-osmosis drinking water were available ad libitum. All animal procedures were approved by the National Institutes of Health Animals Care and Use Committee and were performed in accordance with an approved National Institute of Environmental Health Sciences animal study proposal.

### 4.2. Immunofluorescence, Tunel Assay, and Histological Analyses

Ovaries were collected and fixed in 4% paraformaldehyde overnight at 4 °C. Immunofluorescence experiments were performed on either 5 μm paraffin sections or 10 μm cryosections of 4.5-month and 15-month ovaries. For paraffin sections, slides were dewaxed and rehydrated in a decreasing gradient of alcohol, followed by treatment with 0.1 mM citrate-based Antigen Unmasking Solution (Vector Laboratories, Newark, CA, USA). For both paraffin and cryosections, the slides were blocked in a blocking buffer (5% donkey serum/0.1% Triton X-100 in PBS) for 1 h at room temperature and incubated overnight at 4 °C with the primary antibodies diluted in the blocking buffer. The following day, the slides were washed three times with 0.1% Triton X-100 in PBS and incubated for 1 h at room temperature with the secondary antibodies diluted in the blocking buffer. The 4.5-month samples were then washed and counterstained with DAPI (4′,6-diamidino-2-phenylindole). To alleviate the high levels of autofluorescence in the 15-month-old ovaries due to lipofuscin accumulation [45], the samples were first quenched using the Vector^®^ TrueVIEW^®^ Autofluorescence Quenching Kit (Vector Laboratories, Newark, CA, USA) and the TrueBlack^®^ Lipofuscin Autofluorescence Quencher (Biotium, Hayward, CA, USA) following manufacturers’ instructions. The samples were mounted in ProLong Diamond Antifade Mountant (Thermo Fisher, Waltham, MA, USA). The following antibodies were used: GFP (1:300; ab13970, Abcam, Cambridge, MA, USA), FOXL2 (1:200; NB100-1277, Novus Biologicals, Littleton, CO, USA), CK8 (1:300; ab59400, Abcam), KI-67 (1:300; ab15580, Abcam), pan-LAMININ raised against laminin purified from the basement membrane of Englebreth Holm-Swarm (EHS) mouse sarcoma (1:300; Sigma L9393, MilliporeSigma, Burlington, MA, USA), and AMH (1:500; sc-6886, Santa Cruz Biotechnology, Santa Cruz, CA, USA).

Cell death was analyzed using the Roche In-Situ Cell Death Detection Kit Fluorescein (MilliporeSigma, Burlington, MA, USA). Briefly, the paraffin slides were subjected to an immunofluorescence protocol for FOXL2 and KI-67 as described above. After the secondary antibody washes, the slides were incubated in the TUNEL labeling solution for 1 h at 37 °C, washed in PBS, counterstained in DAPI and mounted. Sections for immunofluorescence and TUNEL assay were imaged under a Leica DMI4000 confocal microscope (Leica Microsystems Inc., Buffalo Grove, IL, USA). For all immunofluorescence or TUNEL experiments, at least 3 independent biological replicates were analyzed. Histological analysis was performed on 5 μm paraffin sections stained with hematoxylin and eosin. The slides were scanned using Aperio ScanScope XT Scanner (Aperio Technologies, Inc., Vista, CA, USA). 

### 4.3. RNA Extraction and Real-time PCR Analysis

Control and KO ovaries were collected at 4.5 months on the day of ovulation (control: *n*  =  7 biological replicates; KO: *n*  =  8 biological replicates). Total RNA was isolated from ovaries using the Qiagen RNeasy Mini Kit (Qiagen, Germantown, MD, USA) following the manufacturer’s instructions. RNA quality and concentration were determined using the Nanodrop 2000c (Thermo Fisher Scientific, Waltham, MA, USA). cDNA synthesis was performed with 1 μg of total RNA, random hexamers, and the Superscript II cDNA synthesis system (Invitrogen Corp., Waltham, MA, USA). Gene expression was analyzed by real-time PCR with the Bio-Rad CFX96 Real-Time PCR Detection system (Bio-Rad, Hercules, CA, USA). The following probes were used to quantify gene expression: *Runx1* (Fw: GCAGGCAACGATGAAAACTACT; Rv:GCAACTTGTGGCGGATTTGTA); *Gapdh* (Fw: TTCACCACCATGGAGAAGGC; Rv:GGCATGGACTGTGGTCATGA). 

### 4.4. RNA Sequencing

RNA sequencing (RNA-seq) was performed on 4.5-month-old control and *Runx1* KO ovaries using 1 μg of total RNA isolated as described above (6 biological replicates/genotype). RNA-seq was performed by Active Motif (Carlsbad, CA, USA). Libraries were sequenced on Illumina NextSeq 500 as paired-end 42 bp sequencing reads. Analysis was performed in R version 3.3.3. The samples were aligned to the mouse mm10 reference genome using the STAR RNA-seq aligner (v2.5.2b) with default parameters. The number of fragments overlapping predefined genes of interest were counted using featureCounts in Subread software package (v1.5.2). The gene annotations were obtained from the NCBI RefSeq database and then adapted to create a set of disjoint exons for each gene. Normalization and differential analysis to identify statistically significant differentially expressed genes (DEGs) were performed using DESeq2 (v1.14.1) with an FDR of <0.1 (Appendix A). The list of differentially expressed genes was analyzed for pathways and gene ontology using the online tool EnrichR [46,47,48]. The RNA-Seq data were deposited under the Gene Expression Omnibus (GEO) Accession No. GSE211241.

### 4.5. RNAscope

Samples for RNAscope were fixed in 4% paraformaldehyde overnight at 4 °C and dehydrated in gradients of ethanol. Ovaries were embedded in paraffin, sectioned at 5 μm and mounted on Fisherbrand Superfrost Plus slides (Thermo Fisher Scientific, Waltham, MA, USA). RNAscope was conducted according to RNAscope Multiplex Fluorescent Reagent Kit V2 protocol for FFPE tissue (ACD bio, Newark, CA, USA), with an *Rspo1* (479591-C2) probe. Slides were imaged under a Leica DMI4000 confocal microscope the following day.

### 4.6. Statistical Analyses

For qPCR, statistical analyses were performed using GraphPad prism 9 software (GraphPad Software Inc., San Diego, CA, USA). The normal distribution was determined using the Shapiro–Wilk test. The unpaired Student’s *t*-test was used to test for differences between the 2 groups, with *p* < 0.001 (indicated by ***). Values are presented as mean ± SEM.

## Figures and Tables

**Figure 1 ijms-23-14442-f001:**
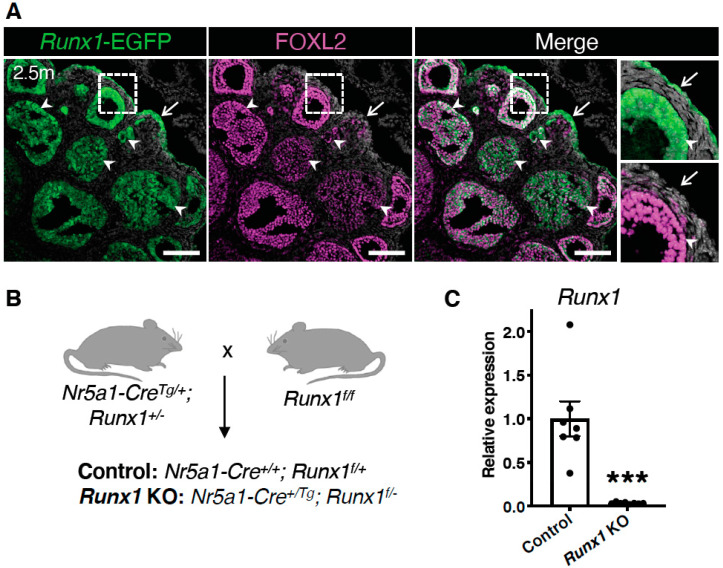
*Runx1* expression in granulosa cells and surface epithelium of adult mouse ovaries. (**A**) Detection of *Runx1*-EGFP and FOXL2 by immunofluorescence in mouse ovaries at 2.5 months of age. Grey color represents nuclei labelled with 4′,6-diamidino-2-phenylindole (DAPI). Arrows point to ovarian surface epithelium. Arrowheads point to granulosa cells. The far-right panels are higher magnifications of outlined areas. Scale bars: 250 μm. (**B**) Mating strategy for the generation of the control and *Runx1* conditional KO mice. All *Nr5a1*+ somatic cells of the ovary were targeted to create a conditional knockout. (**C**) Relative mRNA expression of *Runx1* in control and *Runx1* KO ovaries at 4.5 months (control: *n* = 7; KO: *n* = 8). Values are presented as mean ± SEM; Student t-test, *** *p* < 0.001.

**Figure 2 ijms-23-14442-f002:**
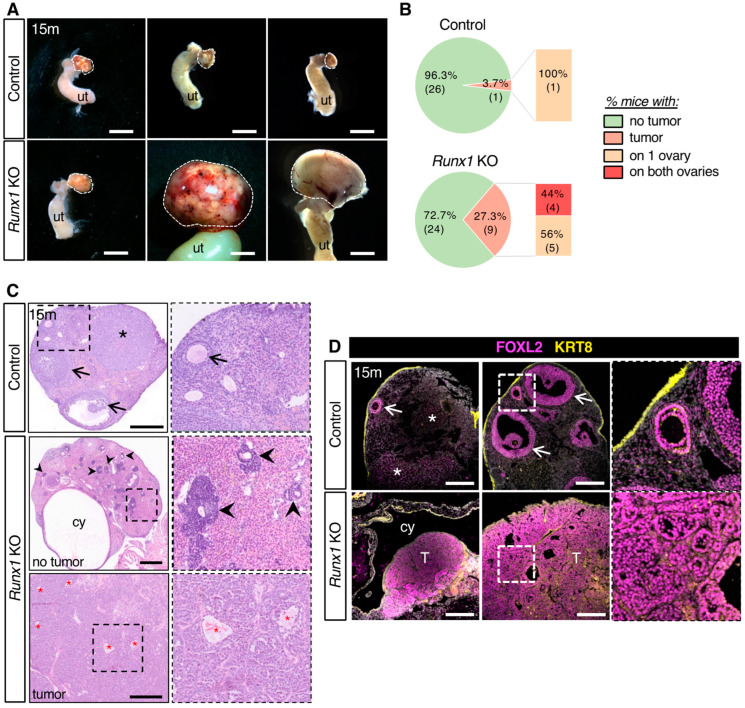
Development of ovarian tumors in aged *Runx1* knockout mice. (**A**) Macroscopic view of control and *Runx1* KO ovaries at 15 months. White dotted lines outline the ovaries. ut represents uterus. Scale bar: 2.5 mm. (**B**) Percentage of mice with macroscopic ovarian tumors in control (*n* = 27 mice) and KO mice (*n* = 33 mice) at 15 months. The numbers in the parenthesis are the actual numbers of animals. (**C**) H&E-stained sections of representative 15-month control and *Runx1* KO ovaries (*n* = 8/genotype). Right panels are higher magnifications of outlined areas. Arrows indicate follicles; arrowheads indicate follicle-like lesions; the black asterisk indicates corpus luteum; Cy represents cyst; red asterisks indicate eosinophilic cavities. Scale bars: 300 μm. (**D**) Immunofluorescence for FOXL2 (magenta) and KRT8 (yellow) in control ovaries and *Runx1* KO ovarian tumors (*n* = 4/genotype). Grey color represents nuclei labelled with DAPI. Right panels are higher magnifications of outlined areas. Arrows indicate follicles; asterisks indicate corpora lutea; cy represents cyst; T represents tumor. Scale bars: 250 μm.

**Figure 3 ijms-23-14442-f003:**
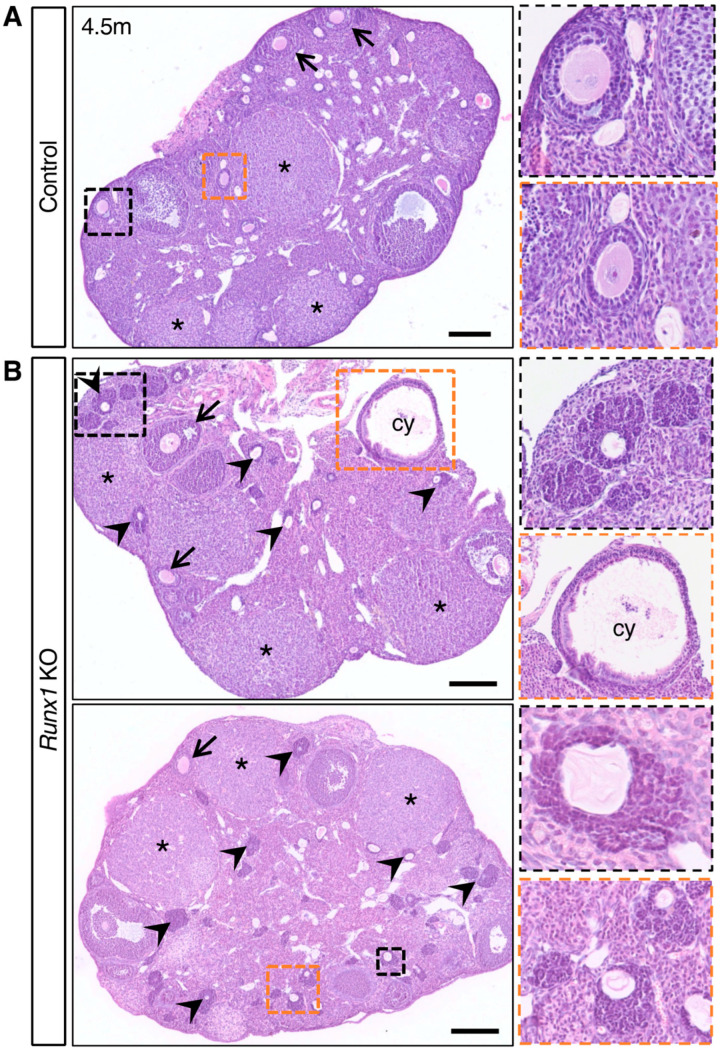
Ovarian defects in young adult *Runx1* knockout mice: H&E-stained sections of representative 4.5-month control (**A**) and *Runx1* KO ovaries (**B**) (*n* = 6/genotype). Black and orange panels on the right are higher magnifications of outlined areas. Arrows indicate normal follicles; arrowheads indicate follicle-like lesions; asterisks indicate corpora lutea; cy represents cyst. Scale bars: 200 μm.

**Figure 4 ijms-23-14442-f004:**
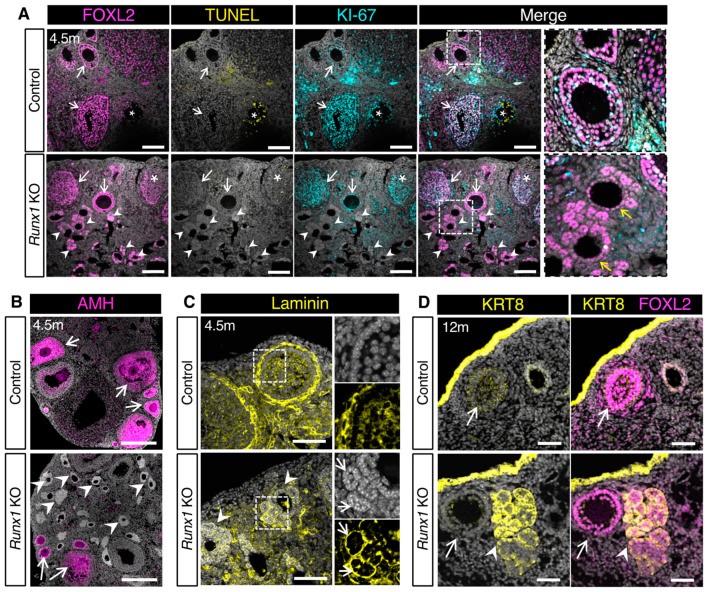
Characterization of the follicle-like lesions in *Runx1* knockout mice. (**A**) Immunofluorescence for granulosa cell marker FOXL2 (magenta), proliferation marker KI-67 (cyan), and TUNEL cell death labeling (yellow) in control and *Runx1* KO ovaries at 4.5 months. The grey color represents DAPI nuclear counterstaining. Right panels are higher magnifications of outlined areas. Arrows point to normal follicles; arrowheads point to abnormal follicles; asterisks indicate atretic follicles with TUNEL+ dying cells; yellow arrows indicate the asymmetric organization of follicle-like lesions with one layer of FOXL2 granulosa cells. Scale bars: 250 μm. (**B**) Immunofluorescence for granulosa cell marker AMH (magenta) with DAPI (grey) in 4.5-month control and *Runx1* KO ovaries. Scale bar: 250 μm. (**C**) Immunofluorescence for basal membrane protein laminin (yellow) with DAPI (grey) in 4.5-month control and *Runx1* KO ovaries. Right panels are single-channel outlined areas with higher magnifications. Arrowheads point to follicle-like lesions; arrows point to rosette-like structures surrounded by laminin within the lesions; scale bar: 50 μm. (**D**) Immunofluorescence for granulosa cell marker FOXL2 (magenta) and epithelial marker KRT8 (yellow) in 12-month control and *Runx1* KO ovaries. Arrows point to normal follicles; arrowheads point to follicle-like lesions. Scale bar: 100 μm; *n* = 4/genotype for all immunofluorescences.

**Figure 5 ijms-23-14442-f005:**
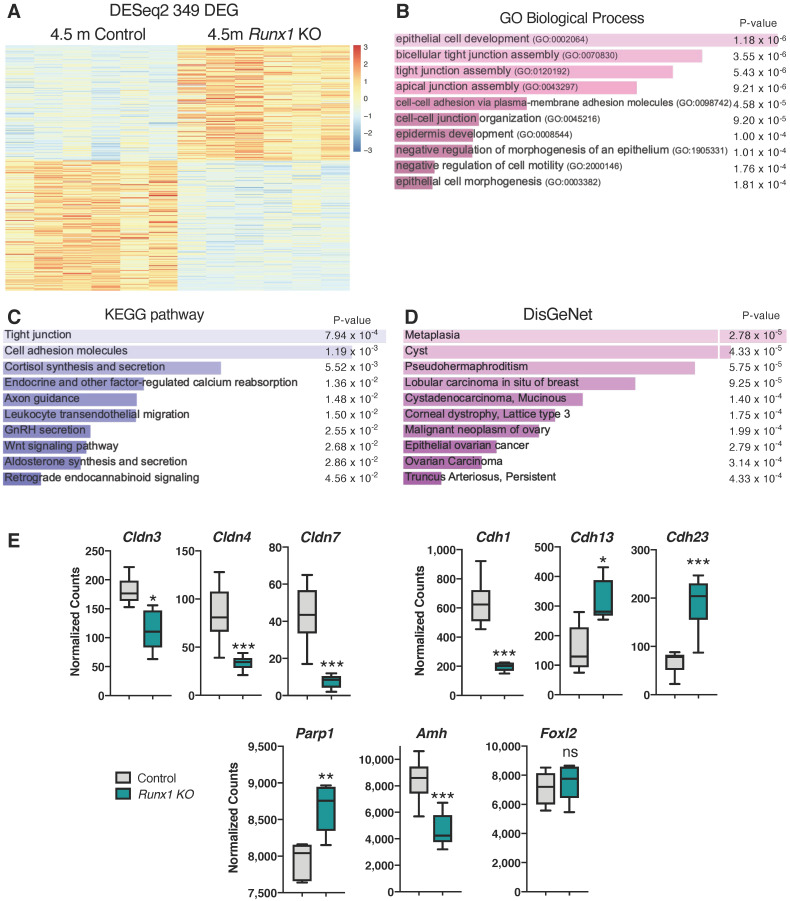
Identification of the differentially expressed genes and pathways in *Runx1* knockout ovaries. (**A**) Heat-map of the 349 differentially expressed genes (DEGs) in *Runx1* KO vs. control ovaries at 4.5 months identified by RNA-seq (*n* = 6/genotype). (**B**) Gene Ontology Biological Process analysis based on DEGs. (**C**) KEGG pathway analysis based on DEGs. (**D**) DisGeNet analysis, which identified human diseases associated with the DEGs. (**E**) Examples of significantly differentially expressed genes in *Runx1* KO vs. control ovaries at 4.5 months. Boxplots represent the DESeq2-normalized counts from RNA-seq data. Values are presented as mean ± SEM (*n* = 6/genotype); ns, not significant; * *p* < 0.05; ** *p* < 0.01; *** *p* < 0.001.

**Figure 6 ijms-23-14442-f006:**
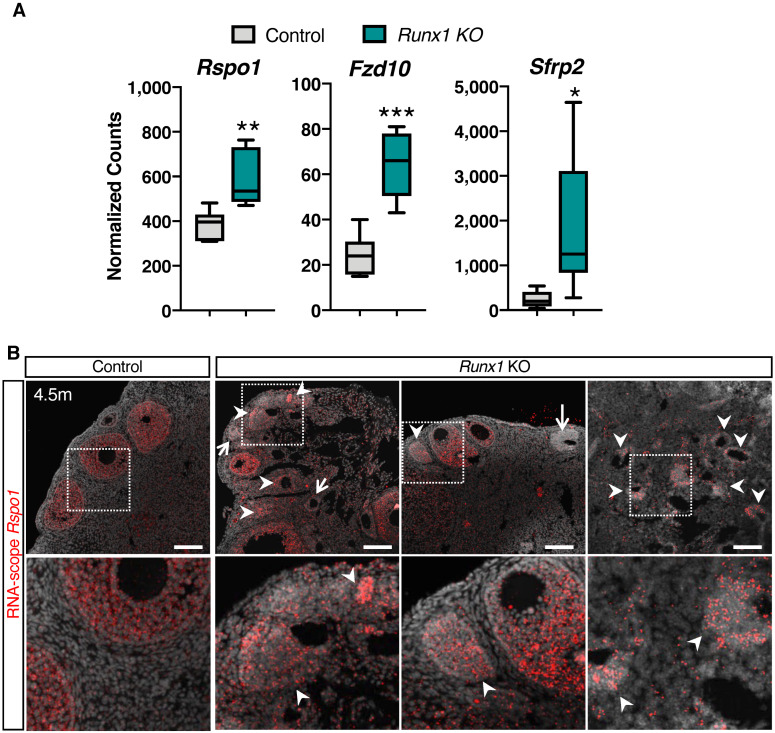
Upregulation of genes of the Wnt signaling pathway in *Runx1* knockout ovaries. (**A**) Examples of significantly differentially expressed genes in *Runx1* KO vs. control ovaries at 4.5 months. Boxplots represent the DESeq2-normalized counts from RNA-seq data. Values are presented as mean ± SEM (*n* = 6/genotype); * *p* < 0.05; ** *p* < 0.01; *** *p* < 0.001. (**B**) RNA-scope for the *Rspo1* probe in control and *Runx1* KO ovaries at 4.5 months (*n* = 4/genotype). Arrowheads point to *Rspo1*+ follicle-like lesions. Arrows point to follicle-like lesions with little to no *Rspo1* expression. Scale bars: 100 μm.

## Data Availability

The RNA-Seq data have been deposited under the Gene Expression Omnibus (GEO) Accession No. GSE211241.

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
