# Peer review of "Loss of Runx1 Induces Granulosa Cell Defects and Development of Ovarian Tumors in the Mouse"

_ijms, 2022, doi:10.3390/ijms232214442_

Round 1

Reviewer 1 Report

This is an interesting study and one which builds on the authors previous work (Nicol et al, Nat Comms, 2019) assessing the role of Runx1 in the murine ovary. Previously the authors assessed fetal development of Sf-1-Cre mediated deletion of Runx1. Here expression in the adult ovaries is assessed, and upon ageing it is ascertained that there is a propensity for Runx1-/- mice to develop ovarian tumours and altered pathology. An RNAseq analysis of young adult mice highlights altered pathways. However, the authors need to be careful not to over-interpret this model as one for the human disease. It is not clear to me how the Sf1-Cre actually models the cell of origin in human ovarian cancer and indeed it is thought (as the authors discuss) that RUNX1 overexpression in human ovarian cancer correlates with poorer survival.

Figure 1: The authors show EGFP expression as a surrogate for RUNX1 expression – could they have performed a RUNX1 IHC or ISH to substantiate these results? In panel C they look for expression in the ovaries and see virtually no expression (RNA) yet SF1 Cre should only delete in the somatic cells. It would be useful to know the levels of FOXL2 in these mice to ascertain if there is an altered population of granulosa cells in the Runx1-KO.

What is the rationale for using Sf1-Cre;Runx1-/fl rather than Sf1-Cre;Runx1fl/fl? In this regard how can the authors be sure there is no effect of the heterozygous while body deletion of Runx1+/-? It also appears from the methods that the control mice are Sf1-Cre;Runx1+/fl which suggests that they are heterozygote and therefore can any haploinsufficiency affects be ruled out?

Was SF1-Cre always as a homozygote transgene – it looks as if this was the case in the methods but it needs to be explicitly said in the text to ensure the variable phenotype of the 15 month adults (27% with tumours) is not due to incomplete penetrance of the Cre.

Was there any abnormality in the other tissues from the SF-1Cre which, for example is also expressed in the red pulp of the spleen

In Figure 2 – how many tumours were tested for FOXL2 expression? Likewise in figure 3 it needs to be shown how many 4.5m mice were tested for altered pathology and in figure 4 – how many different ovaries these result represent.

The authors need to cite the relevant reference for statements made in line 282-284 regards the breast cancer study – I assume they are referring to Chimge et al, Nat Comms 2016.

The RNA seq was carried out on 4.5month ovaries but did the authors carry out any transcriptomic analysis of the ovarian tumours – even by candidate gene RT-PCR if RNAseq was not carried out.

Reviewer 2 Report

In the present study, the AA investigate the effects of Runx1 loss on the ovaries of a conditional Runx1 knockout mouse model. By the deletion of this gene exclusively in the somatic compartment of the ovary, the AA were able to identify early (4.5 m.o.) and late (15 m.o.) effects of Runx1 deficiency. While evidencing some granulosa-specific alteration (quiescent granulosa organized in a rosette-like pattern in abnormal follicle-like lesions) in young KO mice, the AA show the development of granulosa derived tumors in old mice. Moreover, the AA find an alteration in the transcriptome of young KO mice ovaries. The study is well constructed, and the manuscript is well written.

Nevertheless, some major and minor point need to be improved in the current manuscript before acceptance.

Major:

1.     Title: Although the title is clear and straightforward, it needs to be more precise on which mouse strain the AA are using in the study. “The mouse” is a too general description. It has been proved that different mouse strains may have a different response and outcomes to genotype modification and later evaluation in both protein and transcriptome levels. Therefore, it would be useful to the reader to know which kind of strain the AA are using.

2.     Figures: there are a few adjustments to be done on fluorescent images and figures in general.

a.     The AA needs to change the italics font in all the images, since the current one cannot be read clearly and creates mostly confusion (especially in Figure 1B).

b.     Immunofluorescence images seems to be overall oversaturated and the too bright. It seems that the AA are using too much laser power during the image acquisition. In most of the images (GFP and antibody staining) is often impossible to distinguish the signal of single cells.

c.     Add in all the figures (and not only in figure legends) the age of the mouse. Some panels present pictures acquired from mice of different ages. Mixing different ages in the same figure (e.g., figure 4) adds a confounding factor to the interpretation of the results.

3.     Lines 169-170: As the AA state later on during the discussion, the granulosa cells derive from epithelial cells progenitors. Do you think that this Runx1 KO mouse model could push the GCs to undergo some kind of reprogramming and specific dedifferentiation? It would be interesting to check for stemness marker specifically in GCs from 4.5- and 15-months old mice to assess this possibility.

4.     The AA report results for RNA sequencing solely for 4.5-months old mice. Why? Through the whole study, the comparison is made between young and old mice. Which is the reason why the AA decide to sequence only the young counterpart and not to see the effect of Runx1 loss in 15-months old mice? That age is the one with most of the effects of this gene loss.

Furthermore, the results of this bulk RNA sequencing were not validated. This is a huge gap in this part of the manuscript, since it has been proved that the sequencing (especially in bulk) can be biased by a posteriori bioinformatic analyses. Therefore, DEGs need to be validated through RT-PCR in order to prove that this difference is actual and not casual.

5.     Lines 291-292: If this is true, the AA should detect higher amount of RSPO1 mRNA in these lesions. It would be of great value to this paper to prove this hypothesis through a simple RNA in situ hybridization on sections containing both normal follicles and follicle-like lesions. This would strengthen even more your results and discussion.

Minor:

1.     Replace “;” with “,” in the abstract.

2.     Abbreviations: the AA use during the text some abbreviation without without previously mentioning the full name (e.g., abstract “KO”, line 72 “SF1”). I know that some of these abbreviations are widely used, but it may not be clear for some of the readers. Please, write at least once the full name and then keep using the abbreviation during the text.

3.     Line 30: please add a reference at the end of the sentence.

4.     Line 36: I would suggest removing the “or” before genetic factors.

5.     Lines 86-88: As written, this sentence is contradictory. If you mean that there is no difference among ovaries in Runx1 and control, but there is a difference between the two groups, you should explain it better. It would help also adding a table in Fig. 2 or as supplementary material with the mean size with relative error and statistics (among ovaries and between groups).

6.     Line 96: remove “all” from the sentence.

7.     Figure 2: In figure 2A the pictures of the ovaries have been taken at different magnification/distance. Why? Therefore, it is unlikely that the scale bar is the same with different magnification. Please, amend the scale bar with the correct one and state the actual magnification, or show all images at the same distance.

8.     Lines 157, 362 and Figure 4C: Which type of laminin are you targeting? There are several types that have different roles in the ECM composition depending on the ovarian area and surrounding cell types. Please, include this information.

9.     Figure 4D: Why 12-months old and not 4.5 or 15, as in the rest of the manuscript? And why do the AA want to show all the results in ovaries from mice 4.5-months old and only KRT8 in older mice?

10.  Line 209: “Misexpressed”: do the AA mean differentially expressed?

11.  Line 216: “Was found” instead of “found”.

12.  Lines 221-223 and 227-228 are not essential in the result section but should be transferred in the discussion.

13.  Lines 343-345: The number of the ethical approval is missing, as well as quite a few information regarding how the animals were kept (number of light/dark hours, feeding, how many animals per cage, etc.). The lack of these information impairs the reproducibility of the study.

14.  Lines 357-360: Why this method was applied only on 15-months old animal? Autofluorescence is high in the ovary because of the ECM and relative proteins, so the AA should have this issue also with younger animals.

15.  Line 382: delete “: were used”.

16.  RNA Sequencing paragraph in Materials and methods: Are you using R or R studio for the analysis? Please, add this information otherwise it is missing the software used for this analysis.

Reviewer 3 Report

Please find the file attached.

Round 2

Reviewer 2 Report

I would like to thank the authors for their answer and the work behind these revisions. Nowadays, the version of the manuscript has been improved. I recommend the paper for publication after the resolution of some points listed below:

Minor:

1.     Line 14: add “(KO)” after “knockout”.

2.     Figure 2A: I understand the reason behind the larger magnification in the last two Runx1 KO ovaries, but the bar needs to be adjusted accordingly. As they are now, the message delivered to the reader is that all the ovaries are around 150 µm, while the ones in the last two images are around 400-600 µm.

3.     Figure 4D: the age of the mouse (12.5 m.o.) is still missing on this part of the panel. Please, add this info on the image for better clarity.

4.     Lines 169-170: As the AA state later on during the discussion, the granulosa cells derive from epithelial cells progenitors. Do you think that this Runx1 KO mouse model could push the GCs to undergo some kind of reprogramming and specific dedifferentiation? It would be interesting to check for stemness marker specifically in GCs from 4.5- and 15-months old mice to assess this possibility.

Response: Considering the role of Runx1 in cell differentiation, it is possible that loss of Runx1 causes some kind of reprogramming in the cells that normally would express Runx1, such as granulosa and epithelial cells. It is interesting that in our model and other mouse models, some abnormal granulosa cells start expressing epithelial markers. One specific stemness marker picked our interest: LGR5. LGR5 is a marker for stem epithelial cells at the surface of the ovary and is suspected to play a role in serous ovarian cancer. In addition, during neo-natal and postnatal ovarian development in the mouse, granulosa cells that derive from the surface epithelium only arise from LGR5+ precursor cells. Pre-granulosa cells located at the cortex of the developing ovaries express LGR5, and eventually downregulate it as they further differentiate into granulosa cells. It is also interesting to note that LGR5 is a ligand for RSPO1 and is involved in the Wnt signaling. We tested an antibody for LGR5 to see if some cells in the follicle-like lesions expressed it, but unfortunately with no success. This is something we would like to pursue in future studies by focusing specifically on the granulosa cells of controls and KO ovaries.

Thank you for this very interesting and insightful reply. I would suggest the authors to briefly add this information in the discussion or a very short sentence for future perspective of their study.

Author Response

We would like to thank the reviewer for his insightful comments, we have now made the modifications as follow:

  1. Line 14: add “(KO)” after “knockout”.

Response: We have now added “(KO)”.

  1. Figure 2A: I understand the reason behind the larger magnification in the last two Runx1KO ovaries, but the bar needs to be adjusted accordingly. As they are now, the message delivered to the reader is that all the ovaries are around 150 µm, while the ones in the last two images are around 400-600 µm.

Response: We thank the reviewer for catching this error. While all images were taken at the same magnification, the scale bar was incorrect: It was supposed to be 2.5mm instead of 200 um. We have now made the change in the legend. The last two Runx1KO ovaries appear larger because they have macroscopic tumors while the first KO ovary does not. The uterus of second KO image is larger because it was full of fluids, not because of a different magnification.

  1. Figure 4D: the age of the mouse (12.5 m.o.) is still missing on this part of the panel. Please, add this info on the image for better clarity.

Response: we have now updated the figures to show the age of the mouse.

  1. “Lines 169-170: As the AA state later on during the discussion, the granulosa cells derive from epithelial cells progenitors. Do you think that this Runx1 KO mouse model could push the GCs to undergo some kind of reprogramming and specific dedifferentiation? It would be interesting to check for stemness marker specifically in GCs from 4.5- and 15-months old mice to assess this possibility.

Response: Considering the role of Runx1 in cell differentiation, it is possible that loss of Runx1 causes some kind of reprogramming in the cells that normally would express Runx1, such as granulosa and epithelial cells. It is interesting that in our model and other mouse models, some abnormal granulosa cells start expressing epithelial markers. One specific stemness marker picked our interest: LGR5. LGR5 is a marker for stem epithelial cells at the surface of the ovary and is suspected to play a role in serous ovarian cancer. In addition, during neo-natal and postnatal ovarian development in the mouse, granulosa cells that derive from the surface epithelium only arise from LGR5+ precursor cells. Pre-granulosa cells located at the cortex of the developing ovaries express LGR5, and eventually downregulate it as they further differentiate into granulosa cells. It is also interesting to note that LGR5 is a ligand for RSPO1 and is involved in the Wnt signaling. We tested an antibody for LGR5 to see if some cells in the follicle-like lesions expressed it, but unfortunately with no success. This is something we would like to pursue in future studies by focusing specifically on the granulosa cells of controls and KO ovaries.”

Thank you for this very interesting and insightful reply. I would suggest the authors to briefly add this information in the discussion or a very short sentence for future perspective of their study.

Response 2: We have now added this information in the discussion lines 457-467.